# Horizontal and vertical movements of starry smooth-hound *Mustelus asterias* in the northeast Atlantic

Christopher A. Griffiths [1,2]*, Serena R. Wright[1], Joana F. Silva[1], Jim R. Ellis[1], David A. Righton[1], Sophy R. McCully Phillips[1]

1 Centre for Environment, Fisheries and Aquaculture Science (Cefas), Lowestoft, Suffolk, United Kingdom,
2 Department of Animal and Plant Sciences, University of Sheffield, Sheffield, United Kingdom

* c.a.griffiths@sheffield.ac.uk

**Data Availability Statement:** The tagging data collected and described in this paper are freely available via the Cefas Data Hub (https://doi.org/10.14466/CefasDataHub.113).

## Abstract

Commercial landings of starry smooth-hound *Mustelus asterias* in northern European seas are increasing, whilst our knowledge of their ecology, behaviour and population structure remains limited. *M. asterias* is a widely distributed demersal shark, occupying the waters of the southern North Sea and Irish Sea in the north, to at least the southern Bay of Biscay in the south, and is seasonally abundant in UK waters. There are no species-specific management measures for the northeast Atlantic stock, and the complexity of its population structure is not yet fully understood. To address this issue, we deployed both mark-recapture and electronic tags on *M. asterias* to gain novel insights into its horizontal and vertical movements. Our data suggest that the habitat use of *M. asterias* changes on a seasonal basis, with associated changes in geographical distribution, depth utilisation and experienced temperature. We report the first direct evidence of philopatry for this species, and also provide initial evidence of sex-biased dispersal and potential metapopulation-like stock structuring either side of the UK continental shelf. Investigations of finer-scale vertical movements revealed clear diel variation in vertical activity. The illustrated patterns of seasonal space-use and behaviour will provide important information to support the stock assessment process and will help inform any future management options.

## Introduction

The more we learn about the movement, spatial distribution and stock structure of commercially important fish species, the more it becomes clear that some nominal stock units don't always capture the underlying dynamics of the population [1–4]. For instance, in the waters surrounding the British Isles, numerous species (including Atlantic cod *Gadus morhua*, European seabass *Dicentrarchus labrax* and European plaice *Pleuronectes platessa*) have been shown to exhibit metapopulation-like stock structures made up of several semi-discrete sub-populations [5–7]. These sub-populations often converge on particular locations (e.g. for spawning or foraging) and are subsequently dispersed at other times of the year [5–7]. Consequently, such sub-populations may demonstrate unique space-use patterns [4], display

**Funding:** This work was funded by the European Maritime and Fisheries Fund project ENG 1395, Defra and Cefas' science development scheme. The funders provided support for consumables, field work and staff time for all authors (CAG, SRW, JFS, JRE, DAR and SMP), but did not have any additional role in the study design, data collection and analysis, decision to publish, or preparation of the manuscript. The specific roles of these authors are articulated in the 'author contributions' section.

**Competing interests:** Several of the authors are/ were employed by Cefas, however, we confirm that this does not alter our adherence to PLOS ONE policies on sharing data and materials.

differing fine-scale foraging and spawning strategies [8–10] and exhibit variable rates of growth and maturation [11]. Moreover, individuals from one sub-population may experience a very different set of environmental variables or anthropogenic pressures than those of other sub-populations [10, 12]. Despite such variation, metapopulation-like stock structures are rarely addressed in assessment models [1–4] and, by not considering them, we may be over-looking a clear need for regional management measures that more appropriately reflect the underlying stock structure of the species we're trying to protect.

The problem of stock structure is particularly apparent in emergent fisheries, where information about the long-term movement patterns and behaviour of a species are often lacking. One such species is the starry smooth-hound *Mustelus asterias*, which is a medium-sized (<140 cm total length, $L_T$; [13]) triakid shark with a widespread spatial distribution [14–16]. Currently the ICES Working Group for Elasmobranch Fishes (WGEF) considers there to be a single stock of *M. asterias* in the northeast Atlantic, stretching from the southern North Sea and Irish Sea in the north, to a southern limit that remains largely unknown [17]. This uncertainty is mainly driven by species identification issues, as morphological similarities between *M. asterias* and the common smooth-hound *Mustelus mustelus* (a separate species that occupies the warmer waters of the central eastern Atlantic, Mediterranean Sea [14, 15] and Black Sea [16]) have confounded both fishery-independent trawl surveys and commercial catch data [18]. This misidentification may have led to an over-estimation of the stock's spatial range for much of the 20[th] century.

Within UK and adjacent waters, including the southern North Sea and English Channel, *M. asterias* may be seasonally abundant [18] and reported landings of the species by UK fishing vessels have notably increased over the last 18 years [17]. Such increases in landings have been linked to a growing market demand for the species as a viable alternative to other catch restricted shark species, such as spurdog *Squalus acanthias*. Further, fishery-independent surveys clearly show that the relative abundance of *M. asterias* has increased [17], a trend that likely increases its availability and catchability to both commercial and recreational fishers. Currently there are no catch limits or other species-specific management measures in place relating to *M. asterias* [17] and the potential drivers of an increasing stock size at the northern extent of its spatial range remains largely unanswered.

Previous work has demonstrated that *M. asterias* undertakes a circannual migration, shifting seasonally from a summer distribution in the southern North Sea and eastern English Channel to an overwintering distribution in the western English Channel, Celtic Sea and Bay of Biscay [19, 20]. These migrations could be indicative of philopatry, a behaviour where individuals tend to return to a particular area on an annual basis. In other commercial species, for example Atlantic cod, philopatric behaviour has been linked to the formation of sub-populations, that either mix with conspecifics on seasonal grounds or remain in isolation for much of the annual cycle [21, 22]. In the mark-recapture work of Brevé et al. [20], the authors concluded that there may be 'two (or more) populations of starry smooth-hound in the northeast Atlantic Ocean'. This conclusion was not supported by the biological work of McCully Phillips and Ellis [18], where reproductive parameters were shown to be similar, rather than divergent, on both sides of the UK continental shelf. However, if supported by further evidence, for instance via the deployment of electronic tags, the presence of isolated populations or sub-populations that mix on seasonal grounds could have ramifications for our understanding of the *M. asterias* stock and help inform future management options.

Knowledge of vertical movements and depth utilisation can also help assist stock management [23, 24]. Changes in the depth utilisation of fish at varying temporal scales (e.g. diel or seasonal) have been linked to foraging and spawning behaviour [25, 26], predator avoidance [27], migration [28] and orientation [29]. Moreover, fine-scale vertical activity could influence

the vulnerability of fish to capture by commercial and/or survey fishing gears [23, 24, 30]. For instance, electronic tagging studies of *S. acanthias* have demonstrated that bouts of diel vertical activity may reduce the likelihood of capture in bottom trawls during daylight surveys [31, 32], a finding that could impact estimates of stock size [23, 24]. Despite this, very little is currently known about the vertical movements and depth utilisation of *M. asterias*. Electronic tagging of the species could therefore provide important insights into fundamental ecology and help inform both stock assessment and management.

Given current unknowns surrounding the vertical movements of *M. asterias* and the uncertainties relating to horizontal movements, as well as the increasing abundance of this species in UK waters [18] and the increase in reported landings by UK vessels [17], the aims of this study were three-fold. First, to use electronic data-storage tags, augmented by mark-recapture data, to learn about the horizontal and vertical movements of this species. Second, to provide novel insight on their depth utilisation and thermal habitat. Third, to provide the ecological information necessary to inform potential spatial and/or temporal management measures, should these be deemed necessary in the future to support the sustainable exploitation of the northeast Atlantic stock.

## Materials and methods

Specimens of *M. asterias* were obtained for tagging from either fishery-independent trawl surveys or chartered commercial fishing vessels between November 2003 and April 2019. Given the range of platforms utilised over the time period, fish were caught in a range of different gears, including driftnets, bottom trawls, midwater trammel nets, gillnets and longlines. In general, longlines baited with spider crab *Maja* sp. and shore crab *Carcinus maenas* (as opposed to frozen herring *Clupea harengus*, squid and/or whelk *Buccinum undatum*) coupled with short soak-times (<5 hours), caught larger fish in better condition.

### Mark-recapture tags

Prior to tagging, the total stretched body length ($L_T$; in centimetres) of each *M. asterias* was measured and recorded, as was sex (male or female) and maturity ($M_T$; males only). Tagging undertaken on research vessels allowed for total weight ($W_T$; in grams) to be recorded, although weight data could not be collected on commercial vessels. Condition ($C_T$; i.e. health state) at release was categorised as 'lively' (little evidence of injury, regular body movements) or 'sluggish' (minor injuries and/or limited body movement). To minimise subjectivity, $C_T$ was only assessed by the trained tagger.

Mark-recapture tagging consisted of two tag types: (1) Petersen discs (n = 1238) and (2) rototags (n = 152). Tagging via Petersen discs consisted of two plastic discs which were secured externally by a stainless-steel wire to the anterior base of the first dorsal fin (S1 Fig; tagging methods further detailed in [33–35]). One Petersen disc is yellow in colour and lists a unique identification number, whereas the second is red and provides relevant return information. During tagging, the stainless-steel wire was loaded with the yellow disc, inserted through the anterior part of the first dorsal fin and secured with the red disc. The sharp end of the steel wire was then removed using cutting pliers and the remaining steel was coiled and bent at a 90˚ angle to secure the tag. In comparison, tagging via rototags involved the attachment of a livestock rototag either side of the first dorsal fin using a tagging gun. This method was used in the southern North Sea during 2012 and 2013, however, it's use was halted due to excessive biofouling of tags and potential for fin damage (also highlighted in Brevé et al. [20]).

## Electronic tags

**Tags.** During this study, three archival data storage tags (DSTs) were deployed: the Cefas Technology Limited (CTL) G5 DST (https://www.cefastechnology.co.uk/products/data-storage-tags/g5) with float jacket to maximise returns (S2 Fig), the CTL G5 pop-off DST with float jacket (referred to as 'pDST') and the Cefas G6a+ DST. These DSTs have been deployed on a range of fish species in recent years (e.g. [36, 37]). The data are recorded and stored internally, which requires recapture of the tagged fish or tag recovery prior to data extraction. The geographical and behavioural analyses presented here only consider data extracted from the G5 DST tags, so only these tags are described in detail. Returns from the other two electronic tags (pDSTs and G6a+) are presented as recapture positions only.

G5 tags were programmed to record temperature (˚C) and depth (m) at 120 and 30–60 second intervals, respectively. Varying sampling rates were used to maximise battery life.

**Tagging procedure.** The G5 tag was attached in an almost identical manner to the mark-recapture tagging procedure described above. The G5 tag was attached to the coil at the end of the stainless-steel Petersen disc wire using a small loop of monofilament line and crimp, which was then sealed with self-amalgamating tape to minimise corrosion and premature release (S3 Fig). A refinement was made by the addition of silicone discs, which were placed between the dorsal fin and Petersen disc to minimise the risk of chafing, and the position of the wire was placed slightly lower on the anterior base of the dorsal fin into the musculature to accommodate the additional weight and drag of the tag. The length of the monofilament loop was kept as short as possible to minimise any potential abrasion to the posterior margin of the first dorsal fin.

In keeping with animal welfare, only fish >80 cm $L_T$ were tagged with G5 tags. As with mark-recapture tags, $L_T$ was measured and recorded for each fish, as was sex and $M_T$ for males. $W_T$ was measured and recorded where applicable and condition at release was categorised as lively or sluggish.

**Ethics statement.** All tagging procedures were approved by Cefas' Animal Welfare and Ethical Review Body (AWERB) and licensed by the UK Home Office under the Animals (Scientific Procedures) Act 1986. Tagging was conducted under project licence numbers PPL 70/7734 and PPL 9DCB3674 (both titled "Fish Movements and Behaviour") by trained and competent personal licence holders.

**Geographic location.** The daily movements of each *M. asterias* tagged with electronic tags were geographically reconstructed using an adapted version of the tidal geolocation model of Pedersen and colleagues [38]. In brief, the geolocation model used a novel Fokker-Planck based method to combine the geolocation technique of Metcalfe and Arnold [39–41] with a two-state hidden Markov model (HMM), such that an individual's daily location *d* was modelled conditionally on its previous location (*d-1*), its inferred behavioural state $d_s$, where behaviour is defined by a single diffusivity parameter (i.e. the maximum amount of movement permitted in a given day), and the observations made between *d* and *d-1*. In this case, observations consisted of the recorded depth (m; $D_{1,...,n}$) and temperature (˚C; $T_{1,...,n}$), where *n* is the number of measurements made per day (144 in a tag pre-programmed to record every 10 minutes), and any hydrostatic (tidal) data which are derived from the sinusoidal pressure cycle recorded in the depth data when a fish is at rest on the seafloor. The output of the model is a nonparametric probability distribution of geographical position from which a most probable location, for each day at liberty, and a most probable movement path can be estimated [38]. The error associated with each geographical position will vary on a daily basis depending on the certainty in the positional estimate, for example, whether or not a tidal signal was present in the depth data. For further details we refer the reader to [38, 42].

Prior to geographical reconstruction, each fish's depth and temperature measurements were visually assessed and down sampled to a 10-minute resolution. Time often elapses between the date of tag removal from the fish (either from tag beaching or active removal) and the date of data extraction (when the tag stops recording). To account for this, days when tags were floating at the surface were removed where appropriate. Only electronic tags recovered with sufficient data were considered for geographical reconstruction.

### Analysis of fish movement

Analyses and visualisations of *M. asterias* movement were conducted in R [43]. All geographical maps were created using the ggplot2 [44] and mapdata [45] packages. The HMM geolocation model was run in MATLAB [46]. Bathymetry data were sourced from the General Bathymetric Chart of the Ocean online repository [47].

During the analysis a range of different movement metrics/behaviours were calculated for each fish based on tag type (mark-recapture or electronic) and the dimension of movement (horizontal and vertical). Vertical and depth analyses only consider data from electronic tags. Movement metrics/behaviours are calculated as follows:

**Horizontal movement.** For both mark-recapture and electronic tags, the straight-line distance (km; hereby referred to as distance travelled) between release and recapture locations was calculated using the pointDistance function within the raster package [48]. For electronic returns only, a daily distance travelled (km day$^{-1}$) between successive locations was calculated using the same function.

### Vertical speed

Vertical speed (m min$^{-1}$) was calculated by dividing the absolute difference in recorded depth (m) between successive depth observations by the sampling rate (mins). Thus, we consider vertical speed to be representative of both ascents and descents. To investigate both seasonal and diel changes, we summarised vertical speed by calendar month and by day and night. The timing of the day-night shift was calculated using a two-step process. Firstly, sunrise and sunset times for each 24-hour cycle were extracted using the sunrise.set function in the StreamMetabolism package [49]. Secondly, a one-hour buffer was added to both sunrise and sunset to ensure that data were not included for crepuscular periods.

### Proximity to the seabed

To investigate the amount of time *M. asterias* spend in proximity to the seabed, we calculated the proportional time spent within 10 m of the seabed for each day, night and calendar month. It was assumed that the depth of the seabed was equal to the maximum depth reached by an individual during each 24-hour cycle. This approach assumes that individuals do not move into substantially shallower water within a 24-hour cycle.

**Vertical movement behaviour.** Based on the outcome of a two-sample Wilcoxon test, each 24-hour cycle was classified as one of the following vertical movement behaviours: (1) diel vertical migration (DVM), (2) reverse diel vertical migration (rDVM) or (3) no vertical migration (nVM). If depth was significantly deeper during the day than during the night, the fish was considered to be exhibiting DVM. If the reverse was true, the fish was exhibiting rDVM. Alternatively, if the Wilcoxon test yielded a non-significant result (p > 0.05), the day was classified as nVM. To investigate seasonal changes, time spent in each vertical movement behaviour was summarised by calendar month.

## Statistical analysis

Student's t-tests were used to assess whether depth, vertical speed and proximity to the seabed was significantly different during the day than during the night.

## Results

### Releases

A total of 1515 *M. asterias* were tagged and released between November 2003 and October 2019: 1390 with mark-recapture tags (Fig 1; S1 Table) and 125 with electronic tags (S4 Fig; S1 Table). The majority of tagging occurred in the southern North Sea and in the eastern and western English Channel (S2 Table). Tagging most often occurred in Q3 (July-September), a trend that reflects the tendency of *M. asterias* to occupy coastal waters at this time (S3 Table). The average $L_T$ at release was 86 cm (± 15 cm). For those individuals where $W_T$ at release was measured (n = 10), the average $W_T$ was 1.7 kg (± 0.5 kg). One thousand and eighty-four fish were released in a lively condition, 398 in a sluggish condition and 33 had no recorded $C_T$.

### Recaptures

In total, 36 tags were recovered between 2005 and 2019: 18 (1.3% return rate) of these tags were retrieved from fish tagged with mark-recapture tags and 18 (14.4% return rate) from fish tagged with electronic tags (S4 and S5 Tables). Thirty-three of the returned tags were returned with known recapture locations, and 35 with the date of recapture.

Recaptured *M. asterias* remained at liberty for an average of 191 days after tagging (± 181 days). Most of the recaptures (Fig 2) occurred in the coastal waters of the southern North Sea (n = 12) and eastern English Channel (n = 9). Recaptures in these areas peaked in Q3 (S5 Fig; n = 8), closely followed by Q4 (October-December; n = 7) and Q2 (April-June; n = 6). In comparison, recaptures in the western English Channel, Irish Sea, Celtic Sea and Bay of Biscay occurred most often in Q1 (January-March; n = 5) and Q2 (n = 2).

### Horizontal movements

The average distance between release and recapture location of *M. asterias* was 183 km (± 161 km). Two *M. asterias* had recapture locations in the Bay of Biscay, both of which were females (Tag ID 20029 and Tag ID 13937). Tag 20029 had the longest period at liberty (818 days) and travelled the greatest distance between release and recapture location (600 km).

Five recaptured *M. asterias* were tagged and released in the Irish Sea and Celtic Sea during Q4. Of these, two were recaptured in the Bristol Channel, one in the Irish Sea, and two in the Celtic Sea. The average time between release and recapture date of these five fish was 198 days (6.5 months).

When release and recapture locations were analysed by sex (females, n = 22; males, n = 14; S6 Fig), females (203 km ± 174 km; range 1–600 km) were found to have a greater average distance between release and recapture location than males (148 km ± 136 km; range 2–410 km). Individuals released in a lively condition (n = 28) were also found to have a greater average distance between release and recapture location (196 km ± 154 km; range = 2–600 km) than those released in a sluggish condition (n = 8; 135 km ± 154 km, range = 1–400 km; S7 Fig).

### Geographical locations

Of the 18 recaptured electronic tags, only six logged sufficient data to provide geographic reconstructions of migratory movements. The remaining 12 were subject to tag failure (n = 9),

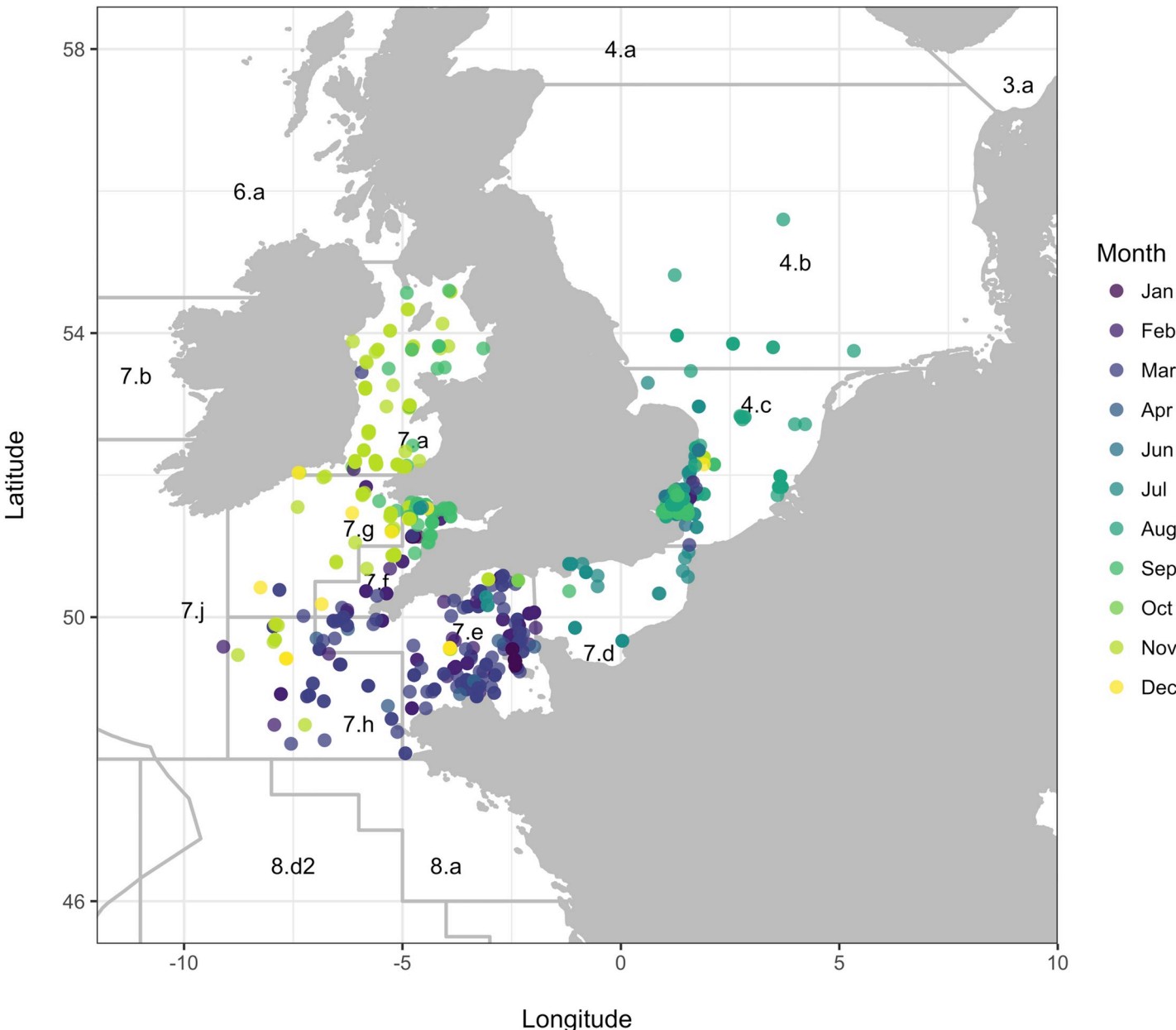

**Fig 1. Release locations of *M. asterias*.** Release locations of *M. asterias* tagged with mark-recapture tags (n = 1390). Points are coloured by month of release. ICES Divisions are labelled and correspond to the following areas: northern North Sea (4.a), central North Sea (4.b), Skagerrak (3.a), southern North Sea (4.c), eastern English Channel (7.d), western English Channel (7.e), Celtic Sea (7.f-h and 7.j), Irish Sea (7.a), west of Scotland (6.a), west of Ireland (7.b) and northern Bay of Biscay (8.a and 8. d2).

had short times at liberty (< 1 month, n = 2) or exceeded the limit of the pressure sensor during deployment (>100 m, n = 1).

Geographic reconstructions revealed two broad movement patterns (Fig 3; S8 Fig). First, one of re-distribution following release in the southern North Sea. Three of the *M.* asterias, all released in Q3, transited down through the eastern English Channel and into the western English Channel and Celtic Sea. These individuals then appeared to reside in these areas for the duration of the Q4 and Q1. Two of the individuals then displayed evidence of a return

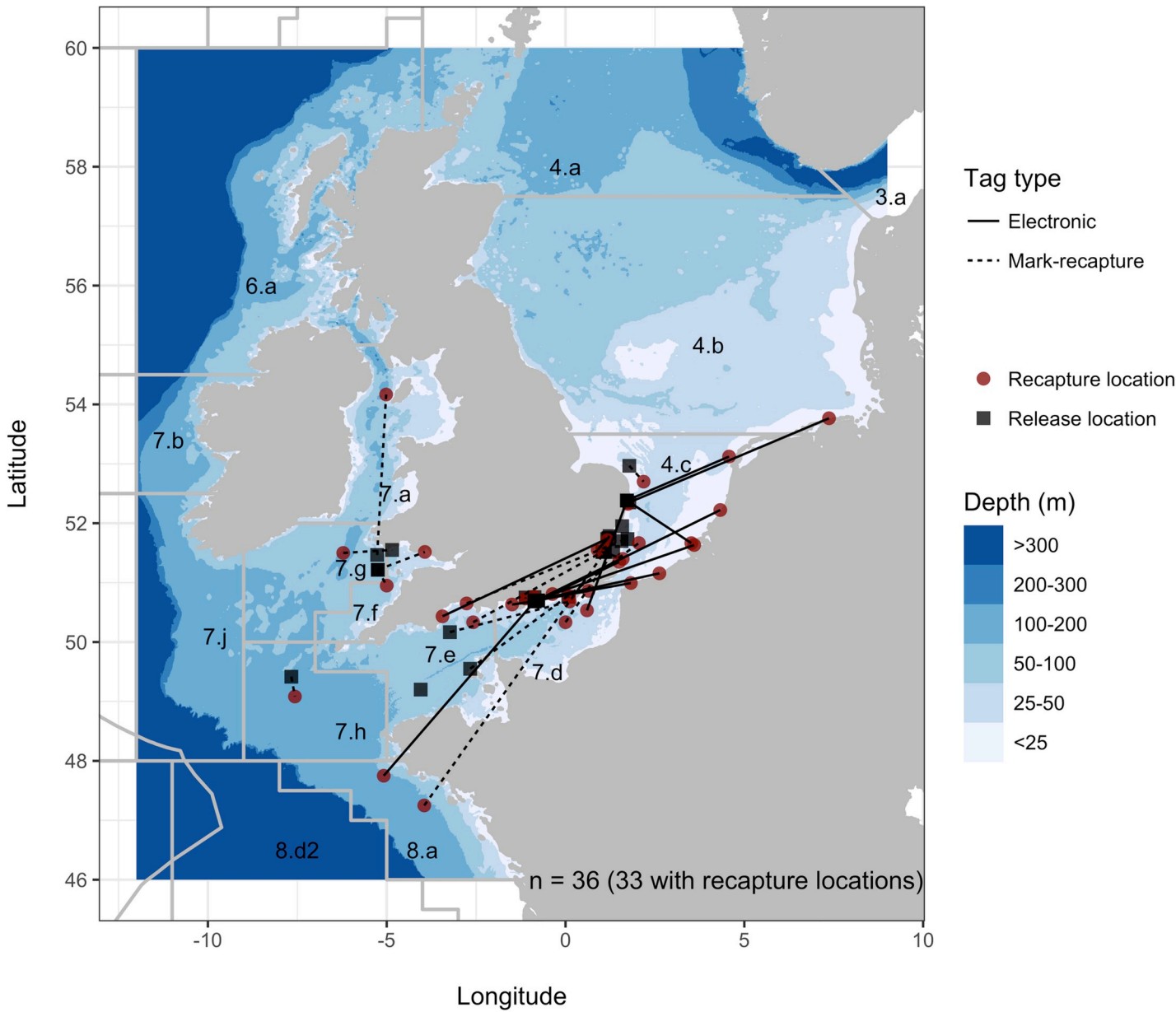

**Fig 2. Release and recapture locations of tagged *M. asterias*.** Release (black squares) and recapture (red circles) locations of *M. asterias* by tag type (mark-recapture, n = 18; electronic, n = 18; S4 and S5 Tables). Lines by tag type represent a straight-line between release and recapture locations and are designed to illustrate migration and dispersal.

migration, with one individual (Tag ID 13733) being recaptured only 34 km from its release location.

The second movement pattern was one of residency following release. Three individual *M. asterias* remained in the general area of their release location for the duration of their time at liberty. Two were tagged in the eastern English Channel and remained in the area, with some displacement into the deeper waters of the western English Channel during Q3 and Q4. The third individual (Tag ID 13693) remained in the southern North Sea. All three individuals were tagged in Q3.

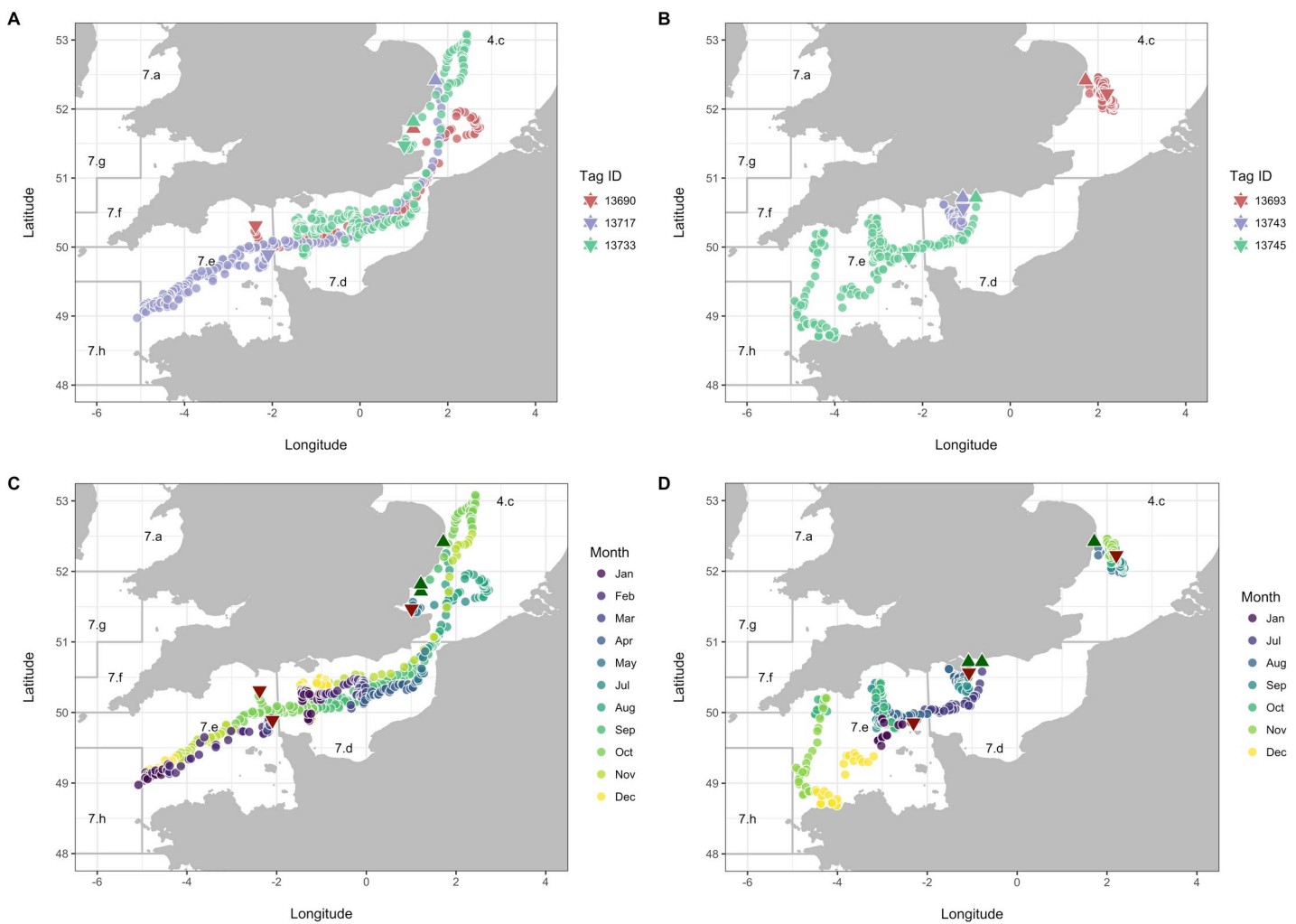

**Fig 3. Movements of *M. asterias* tagged with electronic tags (n = 6).** Daily locations in space and time have been coloured by Tag ID (A and B) and month (C and D; where data are recorded) to illustrate individual variation and seasonality, respectively. Individuals have been split based on movement pattern (migration, A and C; resident, B and D) for illustrative purposes. Release and recapture locations of *M. asterias* are plotted as triangles and inverted triangles, respectively. Each location represents an estimated geographical position per day (24 hours).

## Depth utilisation and temperature

The depth and temperature measurements taken from a single *M. asterias* (Tag ID 13733) are illustrated in Fig 4. In this example it is clear that the individual utilised relatively warm and shallow waters in the months of its release and eventual recapture. In comparison, during Q4 and Q1 the average depth was markedly deeper, and the experienced temperature reduced.

This seasonal shift in depth and temperature was found to be consistent across the six individuals (Fig 5; S6 Table). Mean depth was much shallower during Q2 and Q3 and individuals occupied progressively deeper water throughout Q4. The maximum depth recorded was 118 m on the 12[th] December 2018 (Tag ID 13745). The temperature experienced by *M. asterias* peaked in July and August and was minimal in February and March. The maximum and minimal temperatures recorded were 22°C (Tag 13743; 27[th] July 2018) and 7°C (Tag 13717; 2[nd] March 2018), respectively.

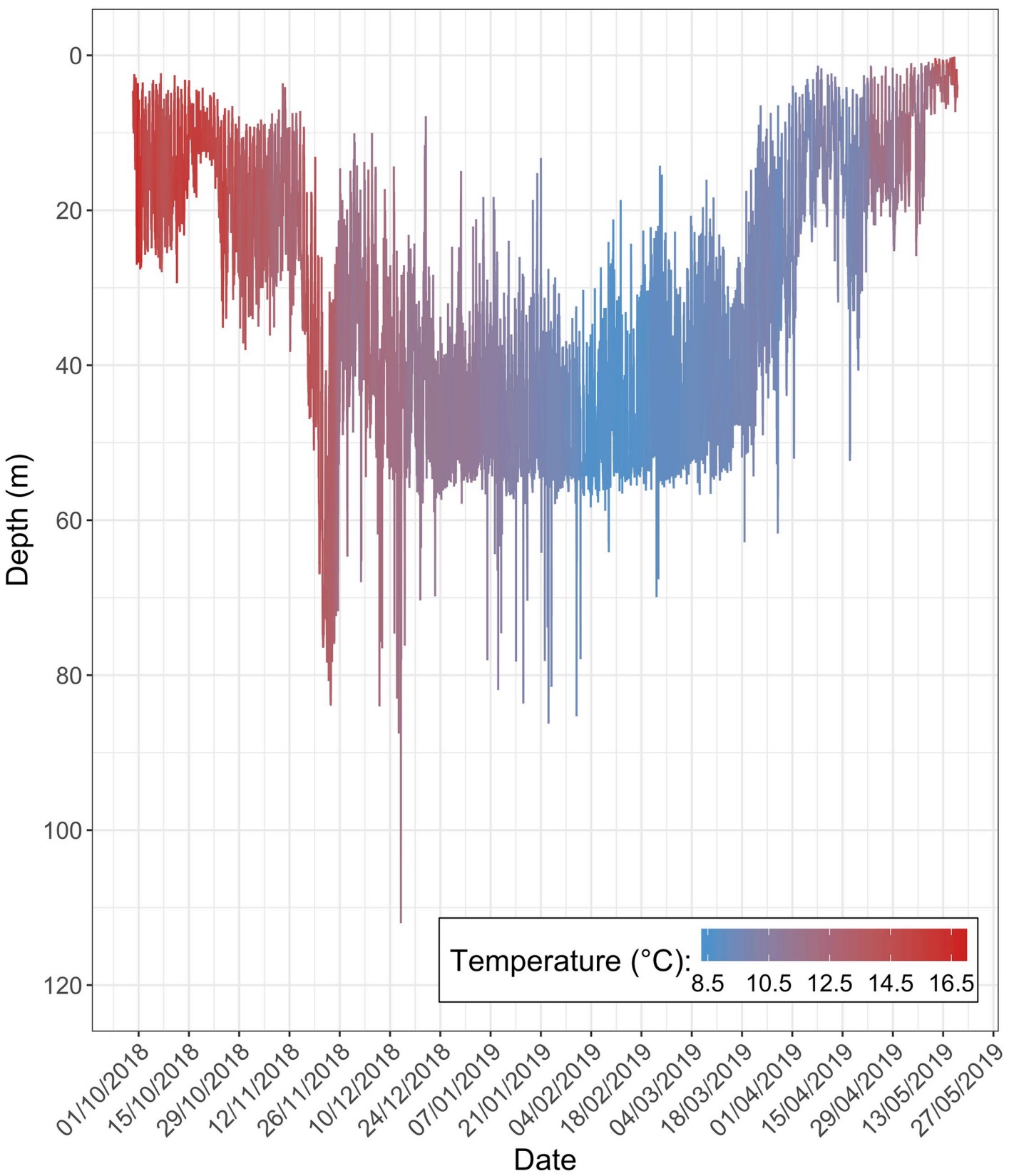

**Fig 4. Depth and temperature measurements taken from Tag 13733.** Depth was measured every 30 seconds and temperature every 120 seconds. This individual was tagged and released on the 29th September 2018 and recaptured on the 16th May 2019 (S5 Table).

Monthly depth was deeper during the day than during the night, albeit non-significantly (t-test: t = 0.83, df = 59.9, p-value = 0.41). In addition, no significant difference was found between the temperature experienced during the day and during the night (t-test: t = 0.009, df = 59.9, p-value = 0.99).

**Vertical speed.** Vertical speed, albeit variable, was found to be higher during Q4, peaking at an average monthly value of 0.27 m min$^{-1}$ in December (Fig 6; panel A). In comparison, vertical speed was reduced in Q2 compared to the rest of the year. Average vertical speeds were much higher during the night (0.27 m min$^{-1}$) than during the day (0.13 m min$^{-1}$; Student's t-test: t = 6.4, df = 47.4, p-value < 0.001).

**Proximity to the seabed.** Time spent in proximity to the assumed seabed remained fairly consistent throughout the year and was highest in January (38%) and May (56%; Fig 6, panel B). As with vertical speed, there were clear diel differences, with *M. asterias* spending proportionally more time in proximity to the assumed seabed (within 10 m) during the day (36.4%) than during the night (20.1%; Student's t-test: t = 4.7, df = 55.8, p-value < 0.001).

**Vertical behaviour.** Of the three vertical movement behaviours considered, DVM was found to be the most dominant, with individuals spending 41–93% of their time exhibiting this type of daily vertical movement behaviour (Fig 7). Some monthly variation existed, for example, in February DVM and rDVM accounted for 41% and 30% of vertical behaviour, respectively. In contrast, 80% of the daily observations in September were characterised as DVM. Despite such variation, DVM appeared to be the dominant vertical movement behaviour, independent of month or quarter.

## Discussion

Past observations [19, 20, 50, 51] have indicated that *M. asterias* is a wide-ranging elasmobranch species, distributed from the southern North Sea and the Irish Sea in the north, to at least the Bay of Biscay in the south; a geographical range supported by the mark-recapture tagging results of this study. Furthermore, our data provide evidence of sex-biased dispersal and potential metapopulation-like stock structuring either side of the UK continental shelf. In addition, data from electronic tags revealed patterns of circannual migration and philopatry, seasonal shifts in geographic location, depth utilisation and temperature, as well as fine-scale vertical movement behaviours. These findings both complement past observations [18–20, 50, 51] and expand on them, providing novel ecological information on a species that remains relatively poorly understood, despite the increase in landings reported by UK vessels in the waters surrounding the British Isles [17, 18].

As with any tagging study, there are limitations that must be acknowledged [52]. Both mark-recapture and electronic tag experiments have returned relatively few tags to date. Thus, whilst we can be confident that the trends described reflect the movements and space use of these individuals, they may not necessarily reflect the spatio-temporal dynamics of the wider population. Further, the data recovered from electronic tags do not cover the full year: there were few observations during Q2 and no depth and temperature measurements were available for June. Hence, we were able to learn about the movements of individual *M. asterias* from July to March but remain uncertain about the rest of the year. Despite this, the fact that physical capture, tagging and release of *M. asterias* is possible during these months does support the hypothesis that individuals are occupying UK waters, including the southern North Sea,

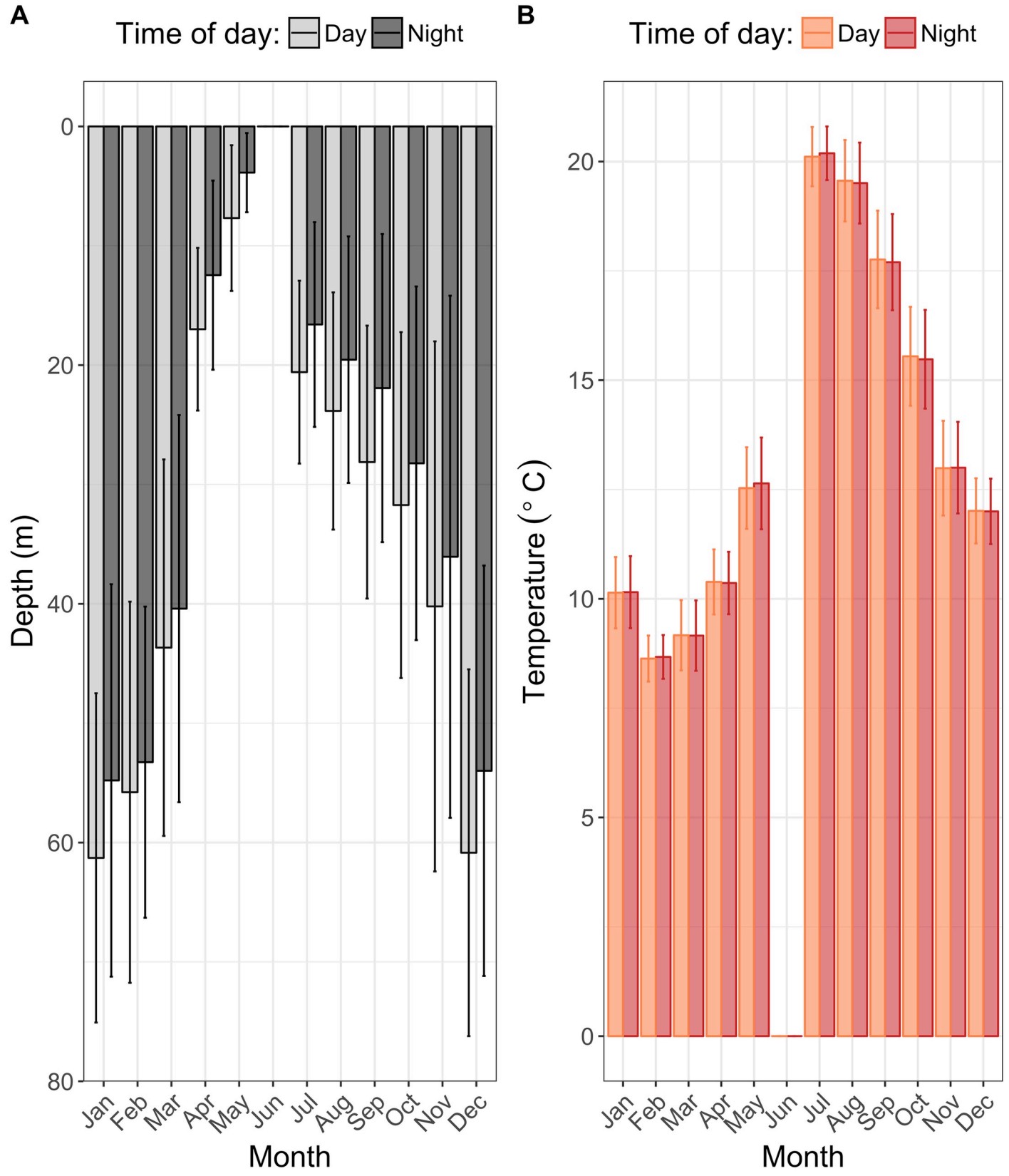

**Fig 5.** Monthly depth (A) and temperature (B) during the day and during the night. Month averages are calculated per fish and per month (where data allow). Error bars represent ± one standard deviation.

eastern English Channel and Irish Sea during this time, and thus provides indirect information on movement and patterns of seasonal space use.

## Sex-biased dispersal

Mark-recapture tag returns indicated that females disperse across a wider geographic range than males, on average 55 km further. Sex-biased dispersal has been reported in a range of elasmobranchs, but typically involves more wide-ranging movements of males [53–55]. Whilst our findings are based on a limited sample size (n = 36), it is noteworthy that recent studies on skates have found that females undertake much longer-distance movements in waters surrounding the British Isles [35] and in the northern Pacific [56]. Similarly, mark-recapture studies by Francis [57] revealed that female rig *Mustelus lenticulatus* around New Zealand travelled further than their male conspecifics. The reasons why females may disperse over larger distances than males could relate to body size (given that female elasmobranchs often attain a larger size than males [18]) and/or reproductive requirements, with females seeking to utilise optimal environments (in terms of temperature and prey availability) to maximise ovarian and embryonic development [58]. Mark-recapture tagging data, however, can only provide information on a minimum distance travelled between release and recapture locations [35, 59], consequently future work that utilises electronic tags may be critical to make more robust inferences on sex-biased dispersal. In the current study, females (63–118 cm $L_T$) were, on average, larger than males (55–91 cm $L_T$) and so further investigations should aim to better disentangle the potential effects of sex, maturity and size on the dispersal of *M. asterias*.

## Horizontal movements

Shifts in the movement and spatial distribution of *M. asterias* appeared to be highly seasonal. During Q3, recaptures of individuals occurred predominantly in the eastern English Channel and southern North Sea. Conversely, during Q1, recaptures occurred more frequently in the western English Channel, Celtic Sea and northern Bay of Biscay. This is corroborated by electronic tagging, as 50% of the individuals were found to move in a southerly direction post-release, transiting down through the southern North Sea and eastern English Channel, and into the western English Channel and Celtic Sea. Such seasonal shifts in movement and spatial distribution of *M. asterias* are analogous to the findings of Brevé et al. [20], and are relatively commonplace in commercial fish species, with individuals often transiting over long distances between different grounds [60–62]. Past observations confirm that *M. asterias* are seasonally abundant in UK waters and often use the warmer coastal waters of the southern North Sea and English Channel as summer pupping grounds [18].

Following residency in the western English Channel, Celtic Sea and Bay of Biscay, there is evidence that individuals return to the southern North Sea and eastern English Channel. Such a return migration was particularly evident from the movements of one individual (Tag 13733) which was recaptured only 34 km from its release location in the southern North Sea. This tendency to return to a particular area on an annual basis is indicative of philopatric behaviour and provides the first direct evidence of philopatry in *M. asterias*. This finding also supports the contention of Brevé et al. [20] regarding potential philopatry in this species and informs our understanding of stock structure and subsequent management options. For instance, in European plaice, philopatric behaviour has been associated with the formation of

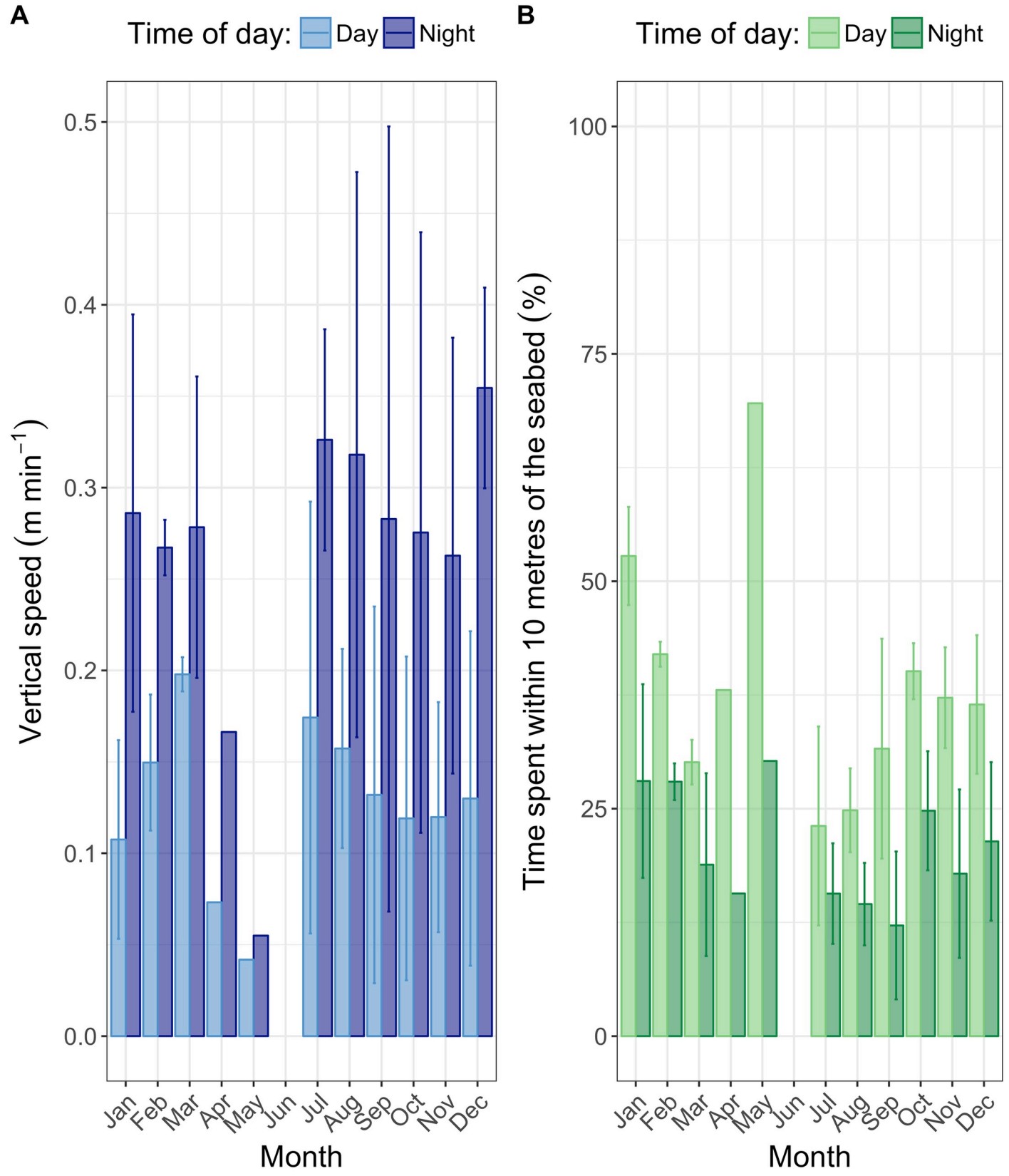

**Fig 6.** Monthly vertical speed (A) and time spent within 10 m of the seabed (B) during the day and during the night. Month averages are calculated per fish and per month (where data allow). Error bars represent ± one standard deviation.

multiple sub-populations that mix on winter spawning grounds but diverge for most of the year [5]. Similarly, in the heavily exploited brown smooth-hound *Mustelus henlei*, genetic analyses have revealed that philopatry in females to natal areas has led to low population connectivity and metapopulation-like stock structuring in the Gulf of California [55].

Mark-recapture tagging data from the Irish Sea suggests that *M. asterias* on the UK's west coast may exhibit different movement patterns to their eastern conspecifics. All four fish that were released in the Celtic Sea (Division 7.g) were recaptured in the same area, or in the Bristol Channel and Irish Sea, and showed no evidence of a southward dispersal. Likewise, one individual (Tag ID 80450) released in the Celtic Sea (Division 7.h) was recaptured in close proximity (38 km) to its release location, despite being at liberty for 11 months. Given the limitations of mark-recapture tagging data (i.e. only knowing the release and recapture locations), it is impossible to know where a fish has travelled during the intervening time period [59]. Whilst more data are certainly needed, the fact that all five fish were tagged and released during Q4, a time when we might expect them to be further south, may suggest that individuals from the west coast exhibit different seasonal space-use patterns. These observations could lend support to the conclusions of Brevé et al. [20] that there may be 'the existence of two (or more) populations of starry smooth-hound in the north-east Atlantic Ocean'. That said, the level of mixing and spatial overlap between the eastern and western components, which may both overwinter in the Celtic Sea and northern Bay of Biscay, remains largely unknown and merits further exploration.

The other movement pattern that was apparent from electronic tagging data was one of residency. Three of the recaptured individuals remained in the general area of their release for the duration of their time at liberty. It could be that these individuals were tagged on their overwintering grounds and had already completed their seasonal migrations. However, release dates in Q3 (S5 Table), landings records [17, 18, 63] and past modelling work [64] suggest otherwise, highlighting that *M. asterias* are abundant in these areas year-round. Furthermore, recent work by McCully Phillips and colleagues [63] demonstrates that European landings of *M. asterias* in 2018 were primarily made from the eastern (33%) and western English Channel (29%), and UK landings (2014–2018) were made throughout the year in these areas. Such trends illustrate that migration into the deeper waters of the western English Channel and Celtic Sea might not be the only behavioural strategy in *M. asterias*, with some individuals opting to overwinter in eastern English Channel and southern North Sea. Given our limited knowledge of the species, residency in the English Channel may be a long-standing characteristic of *M. asterias* ecology. For instance, *Mustelus* was present in ichthyofaunal lists for the English Channel dating back to the late 1800s [50] and early 1900s [19]. Within these ichthyofaunal lists they were considered a common elasmobranch species, but not abundant. Alternatively, the residency of *M. asterias* in the southern North Sea may have become increasingly more common through time due to a northward range expansion of the stock, an overall increase in population size, or a climate driven shift in spatial distribution. For example, several studies have now linked rising sea temperatures and other environmental covariates to both a northward displacement and a deepening of commercially important fish stocks in the North Sea [65–67], as well as northward shifts in some fish distributions to the south-west of the British Isles [68, 69].

## Depth utilisation and temperature

Seasonal shifts in depth utilisation and temperature were also observed. During Q3 *M. asterias* were found to occupy warm shallower waters, whereas during Q4 and Q1 their depth

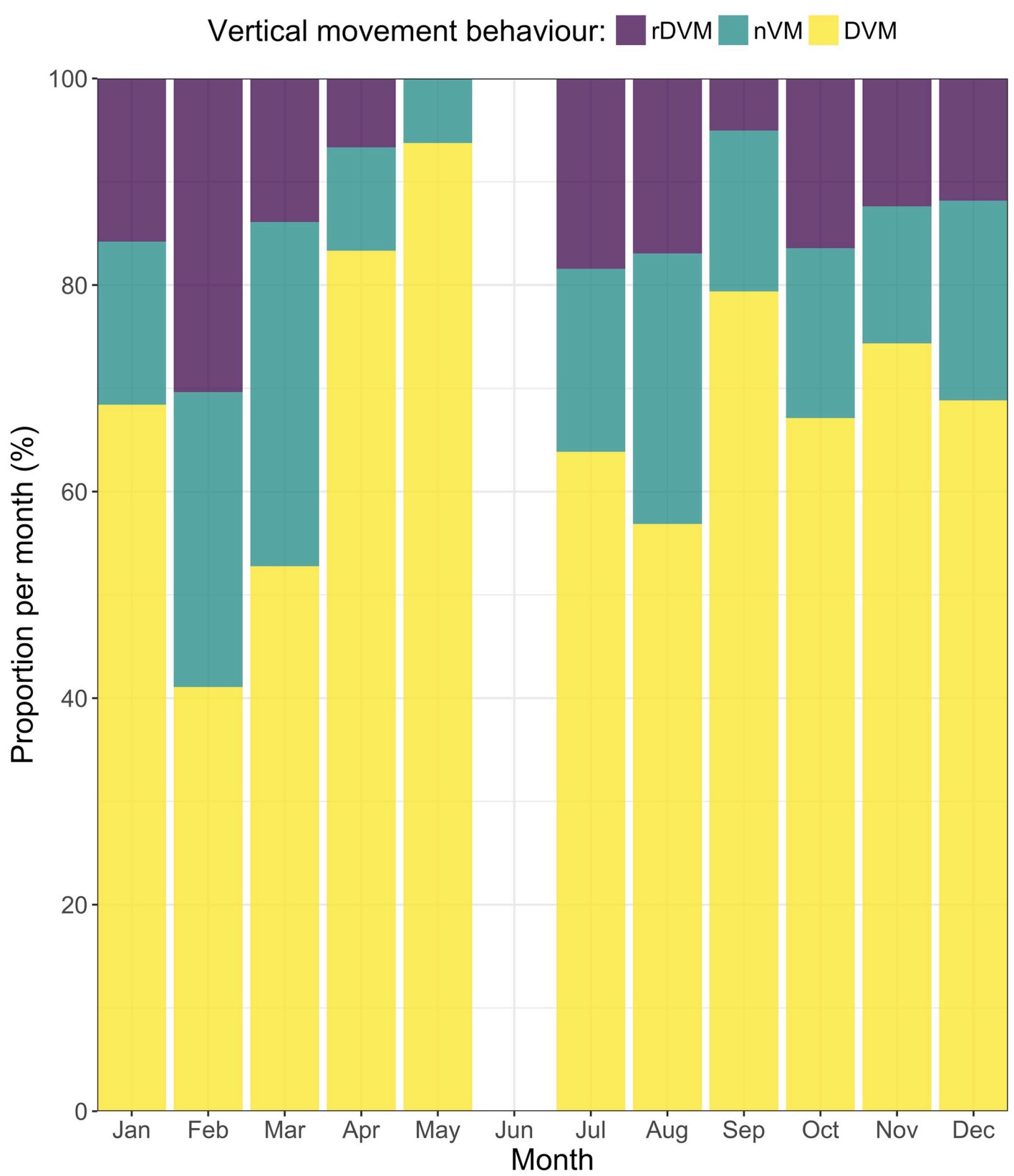

**Fig 7. Proportion of time spent (%) in each vertical movement behaviour per month.** Diel vertical migration (DVM; yellow), reverse diel vertical migration (rDVM; purple) and no vertical migration (nVM; green).

utilisation is markedly deeper and experienced temperatures are reduced. This shift in depth was particularly highlighted in one individual (Tag 13745), which reached a maximum depth of 118 m in mid-December. The use of such depths by *M. asterias* was unexpected prior to this work and to our knowledge provides the first documented case of such depth utilisation in this species. For instance, a recent analysis of trawl survey data suggested that *M. asterias* typically utilise depths <100 m [17].

These shifts in depth utilisation and temperature appeared to be highly seasonal, as were some the observed shifts in geographical location. One potential hypothesis is that the species moves in response to changes in water temperature. Sea surface temperature in the southern North Sea peaks in August [70], and consequently individuals may be residing in these areas until the water temperature reaches some critical threshold which triggers a southward shift. This seasonal shift in location during the winter and spring would then presumably allow individuals to exploit the deeper and warmer waters of the western English Channel [71, 72]. A similar temperature-induced shift has been hypothesised in European seabass, which displays extensive circannual migrations between summer feeding areas in coastal waters to offshore winter spawning grounds [73, 74].

## Vertical movements

Vertical speed (m min$^{-1}$) and time spent in proximity to the seabed displayed clear diel variation. Moreover, DVM appeared to be the dominant vertical movement type independent of month or quarter. During the day, *M. asterias* were less active in the vertical dimension and spent more time on, or in close proximity to the assumed seabed. This then changed during the night, as individuals increased their vertical activity, either moving off the seabed or into shallower waters. Patterns of diel vertical migration appear commonplace in elasmobranchs and have previously been observed in *S. acanthias* [31], white shark *Carcharodon carcharias* [75], scalloped hammerhead *Sphyrna lewini* [76] and bluntnose sixgill shark *Hexanchus griseus* [77], and is often linked to prey searching and foraging behaviour. Dietary studies confirm that *M. asterias* feed almost exclusively (99% index of relative importance) on crustaceans [78]. Consequently, time spent near the seabed could be indicative of visual predation, as individuals feed on their benthic and suprabenthic crustacean prey. An increase in vertical activity during the night may then suggest that individuals are targeting a more active prey type or utilising a differing mode of predation. Alternatively, individuals might be moving into shallower waters and further work is needed to validate seabed association in *M. asterias*. Shifts in prey type seem unlikely, as dietary studies have found minimal evidence of mid-water prey in the diet of *M. asterias* [78]. Instead, the observed changes in vertical movement behaviour could be indicative of resting or sheltering behaviour during the day, as individuals attempt to avoid potential predators. For example, blue shark *Prionace glauca* [79, 80] and common dolphin *Delphinus delphis* [81] are both locally abundant in the western English Channel and could conceivably be potential predators of *M. asterias*.

Diel vertical movement behaviours may also influence the catchability of fish to fishing gears, including commercial fisheries and research surveys [23, 30]. *M. asterias* was found to exhibit lower vertical activities during the day, which would indicate that the (daylight) research trawl surveys that are currently used to assess the species [17] are appropriate. However, if those trawl surveys were to operate over a 24-hour period, then it is possible that catches of *M. asterias* may decrease in those gears, as individuals either move off the seabed, or

utilise shallower waters where their accessibility is reduced. Such behavioural information is rarely considered in stock assessments, but is becoming increasing more recognised [31] and may play a key role in the interpretation of catch-per-unit effort data [23, 24] and in the consideration of management measures.

### Relevance to stock assessment and management

*M. asterias* is of increasing commercial value in northern European seas [17]. Currently, ICES consider a single stock of *M. asterias* in the northeast Atlantic [17], however, the results presented here and those of others [20], suggest that the population may comprise at least two sub-populations. One of these sub-populations spends Q2 and Q3 in the coastal waters of the southern North Sea and English Channel and utilises the deeper waters of the western English Channel, Celtic Sea and northern Bay of Biscay in Q4 and Q1. The other sub-population appears to reside the waters of the Bristol Channel, Celtic Sea and Irish Sea. Despite this, it remains uncertain whether the two sub-populations mix, and future work should prioritise more targeted tagging programmes (e.g. tagging of *M. asterias* in the Bristol Channel and Irish Sea during Q2 and Q3 and in the Celtic Sea in Q4), as well as studies of genetic structure [82], vertebral microchemistry [83] and isotopic signatures [84].

Previous studies link philopatric behaviour to the formation of sub-populations in elasmobranchs [85] and several other commercial fish species [5, 21, 22] and our findings support the hypothesis that philopatry may be driving population structuring in *M. asterias*. It is likely that the observed seasonal shifts in geographical location are linked to parturition, with previous work documenting the presence of gravid females and early-stage juveniles in the shallow coastal waters and bays of the southern North Sea and English Channel [86]. Fidelity to specific areas can limit population connectivity and lead to local stock depletions [87], as well as to an increased vulnerability to human impacts [85, 88, 89]. Conversely, a tendency to repeatedly return to a particular habitat for reproduction does provide the opportunity for affordable and efficient spatial management measures [90, 91], for example marine protected areas, localised no-take zones or regional gear restrictions.

The current assessment for *M. asterias* used by ICES is a stock size indicator derived from daylight trawl surveys in the North Sea (Q1 and Q3) and in the Celtic Sea and Bay of Biscay (Q4) [92]. Our findings demonstrate that the current sampling regime has an appropriate spatio-temporal overlap with the seasonal habitat use of the species. Furthermore, the use of daylight trawls appropriately reflects the described diel variation in vertical activity and inferred catchability. As landings by northern European fishing vessels continue to increase, more emphasis will be placed on sustainable exploitation and the need for a more robust stock assessment. A qualitative stock assessment for *M. asterias* will require knowledge of exploitation levels (past and present), as well as fundamental information of the biology and ecology of the species. Here, we have shown that the species exhibits seasonal variation in geographical location, depth utilisation and experienced temperature. Moreover, we report the first direct evidence of philopatry for this species, and in addition provide evidence of sex-biased dispersal and potential metapopulation-like stock structuring. It is clear that more work is critical to our understanding of *M. asterias*, but the findings presented here go some way to addressing current unknowns.

### Supporting information

**S1 Table. Tagging per year.** Number of *M. asterias* tagged and released with mark-recapture and electronic tags per year.
(PNG)

**S2 Table. Tagging by area.** Number of *M. asterias* tagged and released with mark-recapture and electronic tags by ICES Division.
(PNG)

**S3 Table. Tagging per quarter.** Number of *M. asterias* tagged and released with mark-recapture and electronic tags per quarter.
(PNG)

**S4 Table. Release and recapture information (mark recapture tags).** Release and recapture information for the 18 returned mark-recapture tags.
(PNG)

**S5 Table. Release and recapture information (electronic tags).** Release and recapture information for the 18 returned electronic tags.
(PNG)

**S6 Table. Depth and temperature measurements.** Average depth (m) and temperature (˚C) measurements recorded per month from *M. asterias* tagged with electronic tags. Values are presented per individual and as averages across the six individuals.
(PNG)

**S1 Fig. Petersen disc tagging.** *M. asterias* in the process of being tagged with a mark-recapture Petersen disc. Shown is the yellow disc listing the tags unique identification number.
(PNG)

**S2 Fig. GS and pDST tags.** G5 (top) and pDST (bottom) electronic tags. Both tags were fitted with a float jacket (external orange layer) to maximise return rates. Photos taken from https://www.cefastechnology.co.uk/.
(PNG)

**S3 Fig. *M. asterias* tagged with a G5 electronic tag.**
(PNG)

**S4 Fig. Release locations of electronically tagged *M. asterias*.** Release locations of individual *M. asterias* tagged with electronic tags (n = 125) between January 2017 and October 2019. Points are coloured by month of release. ICES Divisions are labelled and correspond to the following areas: central North Sea (4.b), southern North Sea (4.a), eastern English Channel (7.d), western English Channel (7.e), Celtic Sea (7.f-h and 7.j), Irish Sea (7.a), west of Scotland (6.a), west of Ireland (7.b) and northern Bay of Biscay (8.a and 8.d2).
(PNG)

**S5 Fig. Recapture locations by quarter.** Recapture locations (red) of *M. asterias* by quarter (Q1, n = 8; Q2, n = 6; Q3, n = 10; Q4, n = 9), with ICES Divisions shown.
(PNG)

**S6 Fig. Release and recapture locations by sex.** Release and recapture locations of *M. asterias* by tag type and sex (males, n = 14; females, n = 22), with ICES Divisions shown.
(PNG)

**S7 Fig. Release and recapture locations by condition at release.** Release and recapture locations of *M. asterias* by tag type and condition at release (lively, n = 28; sluggish, n = 8), with ICES Divisions shown.
(PNG)

**S8 Fig. Geographical movements of *M. asterias*.** Movements of *M. asterias* tagged with electronic tags (n = 6). Locations in space and time are coloured by month to illustrate seasonality. No measurements were taken during June. ICES Divisions are labelled where appropriate. (PNG)

## Acknowledgments

We would like to thank the many Cefas staff who have contributed to the project and the fieldwork, in particular Sam Roslyn, Christopher Bird, Matthew Eade, Gary Burt, Louise Cox, Nicola Hampton, as well as the skippers of the vessels who supported this work. We would also like to thank the Home Office License holder (Victoria Bendall), and Ewan Hunter and the anonymous reviewers who provided valuable comments on the manuscript.

## Author Contributions

**Conceptualization:** Sophy R. McCully Phillips.

**Formal analysis:** Christopher A. Griffiths, Serena R. Wright, Sophy R. McCully Phillips.

**Funding acquisition:** Sophy R. McCully Phillips.

**Investigation:** Christopher A. Griffiths, Jim R. Ellis.

**Methodology:** Christopher A. Griffiths, Serena R. Wright, Joana F. Silva.

**Supervision:** Jim R. Ellis, David A. Righton, Sophy R. McCully Phillips.

**Visualization:** Christopher A. Griffiths.

**Writing – original draft:** Christopher A. Griffiths.

**Writing – review & editing:** Christopher A. Griffiths, Serena R. Wright, Joana F. Silva, Jim R. Ellis, David A. Righton, Sophy R. McCully Phillips.

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
