## [Decision Letter · Decision Letter 0]

14 Jul 2020

PONE-D-20-17145

Horizontal and vertical movements of starry-smooth hound *Mustelus asterias* in the Northeast Atlantic

PLOS ONE

Dear Dr. Griffiths,

Thank you for submitting your manuscript to PLOS ONE. After careful consideration, we feel that it has merit but does not fully meet PLOS ONE’s publication criteria as it currently stands. Therefore, we invite you to submit a revised version of the manuscript that addresses the points raised during the review process.

Your  paper has now been revised by two external reviewers. Both reviewers were supportive of the study and encouraged publication, one recommending major revisions and the second with just a very minor editorial revision.

I agree with both referees that this is a good study worth publishing in the scientific literature. As such, I am recommending major revisions. I would highly encourage you to prepare and submit a revised manuscript, taking into account all the comments and corrections provided by the reviewers. I would highlight in particular the main points raised by referee #1.

Thank you, and I am looking forward to receiving a revised version.

We look forward to receiving your revised manuscript.

Kind regards,

Rui Coelho, PhD

Academic Editor

PLOS ONE

Journal Requirements:

2. Thank you for including your competing interests statement; "The authors have declared that no competing interests exist."

We note that one or more of the authors are employed by a commercial company: Centre for Environment, Fisheries and Aquaculture Science (Cefas), Pakefield Road, Lowestoft

Additional Editor Comments (if provided):

Dear Christopher A. Griffiths

Thank you for submitting your paper, which has now been revised by two external reviewers. Both reviewers were supportive of the study and encouraged publication, one recommending major revisions and the second with just a very minor editorial revision.

I agree with both referees that this is a good study worth publishing in the scientific literature. As such, I am recommending major revisions. I would highly encourage you to prepare and submit a revised manuscript, taking into account all the comments and corrections provided by the reviewers. I would highlight in particular the main points raised by referee #1.

Thank you, and I am looking forward to receiving a revised version.

Yours sincerely,

Rui Coelho

Reviewers' comments:

Reviewer's Responses to Questions

**Comments to the Author**

1. Is the manuscript technically sound, and do the data support the conclusions?

Reviewer #1: Yes

Reviewer #2: Yes

2. Has the statistical analysis been performed appropriately and rigorously? 

Reviewer #1: Yes

Reviewer #2: Yes

3. Have the authors made all data underlying the findings in their manuscript fully available?

Reviewer #1: Yes

Reviewer #2: Yes

4. Is the manuscript presented in an intelligible fashion and written in standard English?

Reviewer #1: Yes

Reviewer #2: Yes

5. Review Comments to the Author

Reviewer #1: Horizontal and vertical movements of starry-smooth hound Mustelus asterias in the Northeast Atlantic

This manuscripts uses various tagged methods to describe the horizontal and vertical movements of recently targeted shark species in the UK. The authors do a great job at setting up their study and providing rationale. However, in the methods and results, it becomes difficult to follow. I think this confusion can be fixed by trimming down a lot of the present text that describes the tagging and release information and summarizing all this information in a couple tables. The use of inconsistent terms throughout paragraphs in these two sections contributed to the confusion as well. I do think the low sample size is a slight issue, and although the authors mentioned that, I think too much emphasis has been put on individual behaviors and potential outliers instead of focusing on some of the similarities and trends shared among individual datasets. I also think comparing their results in the context of fishery independent and dependent data where sample sizes are much higher would improve this paper. With the changes suggested below, it definitely seems like the information from this study can complement the current knowledge on this species and also provide some new insight into various behaviors and stock structure.

Introduction

The introduction is well written and clearly supports and provides the rationale for the objectives of this study. Below are a couple suggestions for this section.

L94: I think the word large is a little extreme, may be better to remove it.

L104: The way this sentence is written assumes the authors have already mentioned that knowledge of horizontal and vertical movements are lacking for this species; however, it is not mentioned anywhere in the introduction prior to this. The authors did provide why having that information is important though. Maybe mention vertical movement info is lacking in the same paragraph after that information is presented.

L103. I would argue that based on lines 82-86, horizontal information is not unknown, but that more detailed information is needed.

Methods

I think some reshuffling of this section would be beneficial. The number of individuals tagged with each tag type, in each region, and each time period should be moved to the results section. Leaving this information wouldn’t be that big of a deal if there weren’t so many values reported. There are multiple paragraphs dedicated to these values, which in my opinion would fall under the beginning of the results section.

The detailed explanations of each tagging procedure could probably be shortened if space or word count are issues. The text is difficult to follow as is, especially if the reader is not looking at the pictures in the supplementary figures.

L139: Any reason why Condition is capitalized? Sex, maturity, and weight is not

L156-164: It may be useful to cite studies that used each of these tag types just for reference

L222: should be where not were

L224-242: There should probably be mention of whether the depth and temperature data were summarized or aggregated at all (e.g. by hour), unless data were summarized every 10 mins. If so, please explicitly state that. Also, were data from all three types of electronic tags used for this analysis? What is the error associated with these positions? This is something that should be stated especially since they are estimated positions.

L256: I know it may be obvious, but authors should be clear that for the vertical and seabed analyses they only used data from the electronic tags.

L269: I think this assumption can be dangerous, especially if this species makes a diel migration into shallower waters but remains on the bottom. If this isn’t the case I would state that.

L270: Some may argue that Vertical Speed is also a Vertical movement behavior and thus be included in this section.

L271: Authors should explain why they decided to go with a nonparametric approach.

L274: I think the authors mean reverse not reserve.

Results

The Recaptures, Horizontal movement sections can be substantially reduced. They are also extremely hard to follow as I read from one paragraph to the other because the information being described is very similar among paragraphs. I think having one or two tables describing most of this information would suffice and the authors could simply highlight the most important trends. I don’t think it is necessary to discuss results of individuals (particularly for mark-recapture fish). The authors also partially describe some tagging locations in this text as well which is repetitive from the methods section. I think breaking down the number of recaptures by Quarter is fine but there are times when it is broken down by month which is inconsistent and feels too detailed. As I mentioned further up, it would be best to summarize the tagging and recapture trends in a couple tables and substantially reduce the text.

L281-284: I think it would be best to weave the summary of fish tagged in this section. It would also probably end up being more concise as well.

L291: It might be helpful to define all the Quarters.

L292: Should be where not were

L293: Putting this release information further emphasizes the need to put the other tagging numbers in this section. Also since the authors are comparing characteristics between release and recapture (e.g. dates, days at liberty), it might also be informative to include the average increase in length and weight.

L297: failure should be plural

L301-305: It is a little confusing here because you already talk about recaptures in different quarters on line L291.

L334: There is no description of the tag and recapture symbol in the figure caption.

L338-362: Descriptions on the individual level I think is not necessary. I think describing trends (directional movements, area use, and temporal movements) among these individuals would result in a much clearer section. In addition, the authors provided a figure (Figure 3) that group individuals based on these trends, so it appears to describe the results in the context of those trends would be better. Also there should be a clear description of what each time step position represents, particularly in the figure (every 10 mins as described in the methods? every hour? every day?).

L364-372: Similar to above, this much information on the individual level is not necessary. I think it is okay to show an individual example, but to get into this level of detail takes away from the important trends among individuals described further down. Also further up in the results you use Quarters to describe time and now you are using spring and winter and months instead. Make sure to change this and be consistent throughout.

L379: Again, I don’t think it is necessary to included individual metrics like the depth and the exact date for specific individuals. Does this level of information really help get at the three objectives mentioned at the end of the introduction? You are using seasons here, not Quarters, please change one or other for consistency.

L395: The fact an individual went this fast is cool, but doesn’t help you understand vertical behavior of this shark species as a whole. I understand you have a low sample size, but still think it is important to concentrate the vast majority of text in the results on trends on the species level.

L399: Sounds awkward, please reword.

L391-406: Multiple times authors used some sort of t-test here to compare day and night, but there is no description of this test in the methods section. Is it non-parametric? If not, why is it parametric, while a nonparametric approach was used for geographic location?

L414: There is no acronym for reverse diel vertical migration in figure caption.

Discussion

The discussion was written fairly well. I think the fact that the results were difficult to follow made it a little difficult to follow at least at the beginning. I think it also would improve the discussion if the authors expanded a little more on some of their ideas.

L421: I think this statement is obvious and doesn’t need to be stated here.

L450: Where are female skates known to move further distances? Do all skates species do this?

L450-451: Any speculation or hypothesis why females travel further than males?

L473-478: This is a little confusing because at first you mention individuals that migrate, but then you focus on one particular individual. I think this would be clearer if these trends were articulated better in the results.

L506: I think distributions from this study should be more directly compared to what the landings data suggest. Expanding on this may help.

L516: It is important to cite some studies that found this trend in species for your study area here.

L518-522: This is all summarized results. I don’t have a problem with some repetition between the results and discussion but this is almost a whole paragraph of repeated results.

L531: remove in spatial distribution

L531: I think providing some temperature reference between the southern North Sea and western English Channel would help the reader understand the thermal differences between the two areas. As it stands, latitudinally they don’t appear too different.

L547-551: Split into multiple sentences.

L550: What if they aren’t moving off the bottom but moving to shallower waters to feed on crustaceans, but it looks like they are moving up in the water column?

L562: Again, unless the sharks aren’t moving off the bottom, but just moving to shallower habitats, the trawl would still catch them unless the habitat is too shallow to fish. Is there not trawl data on M. asterias catch at night? I wonder if depth of catch of M. asterias differs between day and night?

L573-575: Would be nice to include something about what type of work would help address this question…acoustic telemetry? Genetic study?

L587: Back to Quarters again, instead of seasons.

Figures

I imagine that if accepted, figures will be uploaded at a higher resolution because right now, it is difficult to read the letters on the figures on the pdf document.

Figure 1. It would be helpful to include the geographic names on the map for readers not from the UK (e.g. Celtic Sea, southern North Sea, etc)

Figure 4. Why not combine the depth and temperature figure into one plot? That is commonly done in many archival studies.

Figure 7. I assume eVM in the figure legend should be nVM.

Supplementary Data

Supplementary Table 2 may be better to just include May as its own row and put 0 for Number of fish tagged.

Reviewer #2: "Horizontal and vertical movements of starry-smooth hound Mustelus asterias in the Northeast Atlantic" is a novel study that provides valuable information on a data-limited species. Information on ecology and species behaviour is essential in fisheries management, therefore this manuscript is a good contribution to fill knowledge gaps on these issues for the starry smooth-hound. The manuscript meets the requirements for publication; results are presented in a straightforward and organized way and it is well written.

Minor comment:

Lines 321, 328, 336: replace "spilt" with "split"

6. PLOS authors have the option to publish the peer review history of their article (what does this mean?). If published, this will include your full peer review and any attached files.

Reviewer #1: No

Reviewer #2: No

---

## [Author Response · Author response to Decision Letter 0]

20 Aug 2020

See response to reviewers document.

---

## [Editor Report · Decision Letter 1]

7 Sep 2020

Horizontal and vertical movements of starry smooth-hound *Mustelus asterias* in the northeast Atlantic

PONE-D-20-17145R1

Dear Dr. Griffiths,

We’re pleased to inform you that your manuscript has been judged scientifically suitable for publication and will be formally accepted for publication once it meets all outstanding technical requirements.

Kind regards,

Rui Coelho, PhD

Academic Editor

PLOS ONE

Additional Editor Comments (optional):

Dear Christopher A. Griffiths,

Thank you for submitting the revised paper.

I believe you and your co-authors have done a good job at addressing and answering all issues raised by the referees. I have no further corrections and believe that the paper can now be accepted for publication.

Yours sincerely,

Rui Coelho
---

## [Editor Report · Acceptance letter]

9 Oct 2020

PONE-D-20-17145R1 

Horizontal and vertical movements of starry smooth-hound *Mustelus asterias* in the northeast Atlantic 

Dear Dr. Griffiths:

I'm pleased to inform you that your manuscript has been deemed suitable for publication in PLOS ONE. Congratulations! Your manuscript is now with our production department. 

Kind regards, 

on behalf of

Dr. Rui Coelho 

Academic Editor

PLOS ONE